# Physicians' knowledge about palliative care in Bangladesh: A cross-sectional study using digital social media platforms

Jheelam Biswas [1,2]*, Palash Chandra Banik [1], Nezamuddin Ahmad [2]

1 Department of Noncommunicable Diseases, Bangladesh University of Health Sciences (BUHS), Dhaka, Bangladesh, 2 Department of Palliative Medicine, Bangabandhu Sheikh Mujib Medical University (BSMMU), Dhaka, Bangladesh

* jheelam.biswas@gmail.com

## Abstract

### Introduction

Palliative care is still a new concept in many developing countries like Bangladesh. Basic knowledge about palliative care is needed for all physicians to identify and provide this care. This study aims to assess the preliminary knowledge level and the misconceptions about this field among physicians.

### Methods

This cross-sectional study was conducted among 479 physicians using a self-administered structured questionnaire adapted from Palliative Care Knowledge Scale (PaCKs) on various digital social media platforms from December 2019 to February 2020. Chi-square, Fisher's extract test, and the Monte Carlo extract test was done to compare the knowledge level with the study subjects' demographic variables.

### Results

An almost equal number of physicians of both genders from four major specialties and their allied branches took part in the study (response rate 23.9%). The majority (71%) of the respondents had an average to an excellent level of knowledge about palliative care, with a median score of 11.0. Although most physicians had average knowledge about the primary goals and general concepts of palliative care, misconceptions are highly prevalent. The commonly present misconceptions were that palliative care discourages patients from consulting other specialties (88.9%), refrains them from taking curative treatments (83.1%), and this care is only for older adults (74.5%), cancer patients (63%), and the last six months of life (56.4%). Age, educational qualifications, and specialties had significant relationships ($P<0.05$) with the level of knowledge.

### Conclusion

Despite having average or above knowledge about palliative care, the physicians' prevailing misconceptions act as a barrier to recognizing the need among the target populations. So,

---

**Data Availability Statement:** All the data relevant to this manuscript can be found at Mendeley Data, DOIs 10.17632/dc2nsmyfr3.1.

**Funding:** The authors received no specific funding for this work.

**Competing interests:** The authors have declared that no competing interests exist.

proper education and awareness among the physicians are necessary to cross this field's barrier and development.

## Background

In recent years, the world has experienced a shift in the disease pattern, and the prevalence of incurable life-limiting diseases is increasing [1]. Palliative care is an approach to prevent and provide relief to the patients' physical, psychosocial, and spiritual sufferings and their families with such illness [2]. Early identification and referral to palliative care have been proven to improve patients' quality of life with terminal illnesses and their families as well as minimize the health care expenditure [3–5]. About 40 million people need palliative care worldwide, but only about 14% are currently receiving it [6]. Although palliative care is not a new concept, access to this care is still inadequate and neglected in developing countries [7, 8].

Several studies have identified various barriers to accessing palliative care, one of which is the lack of proper knowledge and misconceptions among the general populations and health-care providers alike [9–11]. Although palliative care is an integral part of the United States' healthcare system, less than half of the general population knows about it [12]. Most people from developing countries like India, Pakistan, and Nigeria did not even hear of palliative care [13–15]. The situation is slightly better with the health care providers, but still not great. Several studies showed that, though most (62.61%) of the physicians have average knowledge about palliative care's general concepts, they also harbor misconceptions [11, 16, 17]. Many physicians associate palliative care only with pain management [16]. Other general misconceptions about palliative care among physicians are palliative care discourages patients from taking other curative treatment, and it was suitable only for the last three months of life [18].

The field of palliative care in Bangladesh is still new and in the stage of continued development. Approximately 0.6 million patients need palliative care in Bangladesh, but less than 4000 people had received this care until now [19, 20]. Very few studies were done to assess the knowledge level about palliative care in Bangladesh. A study among non-medical students of life science found that one-fourth of the participants are not aware of palliative care [21]. Another study found that only 10% of the practicing Bangladeshi physicians have knowledge about pain management in terminally ill patients [22]. However, no study has yet to provide a clear picture of the knowledge level or the prevailing misconceptions about palliative care among Bangladeshi physicians. As the demand for palliative care is rising, all physicians need to acquire proper knowledge about this field. Now it is high time to explore this knowledge gap.

This paper aims to assess the knowledge level and the misconceptions about palliative care among Bangladeshi physicians to understand the situation better and raise awareness.

## Methods

### Study design and setting

This cross-sectional study was conducted among Bangladeshi physicians of various disciplines throughout the country using a convenient sampling technique. Many Bangladeshi physicians are very active in social media like Facebook, Viber, and WhatsApp and use these platforms to exchange professional views with peers and communicate with general people. This group of physicians was targeted as they were often in touch with other disciplines and general people and easy to contact. Data collection was done from December 2019 to February 2020.

## Sample size and criteria

The estimated sample size was 299 (considering familiarity with palliative care among Bangladeshi nonmedical students 75.2%) [21]. Registered physicians from any discipline and not directly associated with palliative care were included in the study. Initially, 2000 physicians who fulfilled the necessary inclusion criteria showed interest after learning about the study's objectives and purpose, but 479 returned with the complete response (response rate 23.9%).

## Data collection procedure

Data were collected by a self-administered structured questionnaire adapted from Palliative Care Knowledge Scale (PaCKs), a validated tool for measuring preliminary knowledge about palliative care [23]. English version of the questionnaire was used. It contains 13 items about various aspects of palliative care to be answered via "True" or "False" responses. To avoid guessing, we included a third option, "I do not know." This modified version was adapted from a study done by Kozlov E et al. (2017) [12]. A pilot study was done to validate the tool among 50 physicians, which yielded almost the same result. A notice was posted in all known physicians' social media groups simultaneously about the study's objectives and purpose and requested to send e-mail addresses if interested. After checking the details of the interested persons' social media profiles with the respective owners' permission to avoid fake profiles and duplication, the questionnaire was sent to the provided e-mail addresses and requested to return with their responses. Data was collected via the Google form platform, and an informed consent statement was added to the 1st section as a mandatory field.

## Data analysis

The PaCKs knowledge score was calculated using Microsoft Excel 2010 and entered in SPSS version 22.0; editing and logical checking were done and analyzed. Each correct answer was given the score "1" and incorrect responses were scored as "0". The third option, "I do not know" was included to avoid guessing, was also considered and merged with incorrect responses and scored "0". The PaCKs items were divided into two groups: General conceptions about palliative care (item no 1, 2, 3, 9, 10, 12, 13) and Common misconceptions about palliative care (item no 4, 5, 6, 7, 8, 11). We did a descriptive analysis (frequency, percentage, median and interquartile range) for categorical and quantitative variables.

Knowledge level about palliative care was categorized into three categories according to the interquartile ranges and median. The value below the 25th quartile was categorized as poor, the range between the 25th and 75th quartile was categorized as average, and the value above the 75th quartile was categorized as excellent knowledge about palliative care.

Chi-square, Fisher's extract test, and the Monte Carlo extract test was done to compare the knowledge level with the study subjects' demographic variables setting the α level at 0.05.

## Ethical considerations

This study was performed following the Declaration of Helsinki, and no invasive procedures were involved. Verbal permission was taken from the head of the Department of Palliative Medicine, Bangabandhu Sheikh Mujib Medical University, and ethical clearance from the Ethical Review Committee (ERC) of the Center for Noncommunicable diseases Prevention Control Rehabilitation & Research (Approval no: CeNoR/EA/1903, date: 05/10/2019). An informed consent statement was attached at the 1st section of the Google form platform as a mandatory field. Interested participants were allowed to move to the next steps of the questionnaire and submit only after agreeing with the consent statement.

## Results

Almost equal numbers of physicians from both genders took part in the study of mean age 28.9±3.6 years. The majority (78.5%) of the respondents belonged to 20–30 years. Only one-third (34.9%) of the physicians had specialist postgraduate degrees or post-graduation courses. Aside from general practitioners and interns who were almost half (44.1%) of the total respondents, physicians from four primary specialties and their allied branches (Medicine, Surgery, Gynecology and Obstetrics, Basic Medical Sciences, and Public Health) participated in the study. Almost all (96.2%) of the physicians were familiar with the term "palliative care"(Table 1).

The knowledge about palliative care score ranged from as low as 0 to the perfect score of 13, with a median score of 11.0. The proportion of average to excellent knowledge about palliative care was 71% (Fig 1).

Most of the physicians gave correct answers to all the questions regarding general conceptions about palliative care. Most (91.4%) of them recognized palliative care as a team-based approach. They were also aware of the functions and primary goals of palliative care, such as dealing with psychological issues of the patients (76%), improving patients' daily activities (89.6%), addressing the stress of the illness and side effects of other treatments (72.2%), and helping the families to cope (94.2%). Unfortunately, most of the physicians thought the misconceptions about palliative care as accurate. The most prevailing misconceptions were that palliative care discourages patients from consulting other specialties (88.9%) and refrains them

**Table 1. Baseline characteristics of the respondents (n = 479).**

| Variables | n (%) | 95% CI | |
|---|---|---|---|
| | | Lower bound | Upper bound |
| **Sex** | | | |
| Men | 263 (54.9) | 50.4 | 59.4 |
| Women | 216 (45.1) | 40.6 | 49.6 |
| **Age,** *years* | | | |
| *Mean ± SD* | 28.9±3.6 | | |
| 20–30 | 376 (78.5) | 74.8 | 82.2 |
| 31–40 | 98 (20.5) | 16.9 | 24.1 |
| >40 | 5 (1.0) | 0.1 | 1.9 |
| **Qualifications** | | | |
| Graduate (Medical and Dental) | 312(65.1) | 60.8 | 69.4 |
| Postgraduate or in course | 167 (34.9) | 30.6 | 39.2 |
| **Specialties** | | | |
| Basic medical sciences* | 24 (5.0) | 3.0 | 7.0 |
| Medicine and allied branches | 128 (26.7) | 22.7 | 30.7 |
| General practitioners | 136 (28.4) | 24.4 | 32.4 |
| Gynecology and Obstetrics | 17 (5.6) | 3.5 | 7.7 |
| Interns | 75 (15.7) | 12.4 | 19.0 |
| Surgery and allied branches | 57 (11.9) | 9.0 | 14.8 |
| Public health | 32 (6.7) | 4.5 | 8.9 |
| **Familiarity with the term palliative care** | | | |
| Yes | 461 (96.2) | 94.5 | 97.9 |
| No | 18 (3.8) | 2.1 | 5.5 |

*Anatomy, Physiology, Biochemistry, Microbiology, Pathology

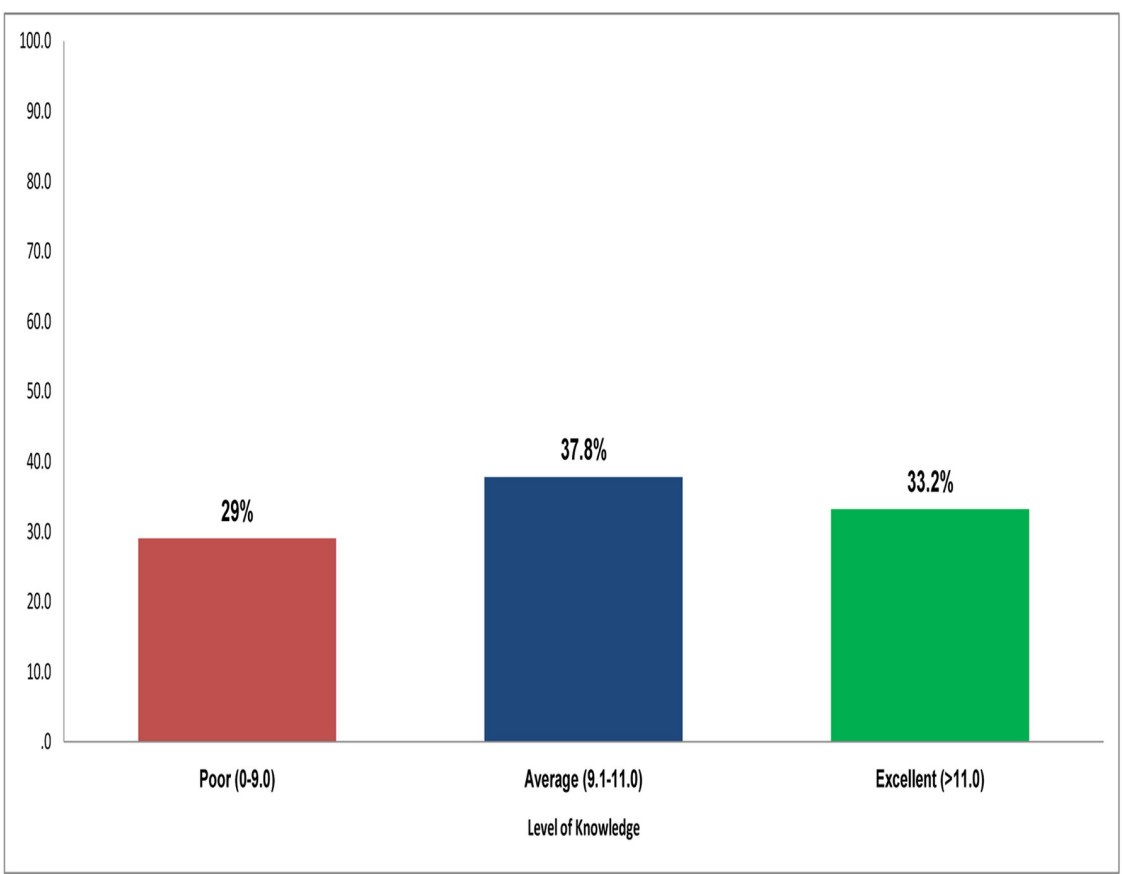

**Fig 1. Levels of knowledge regarding palliative care (n = 479).**

from taking curative treatments (83.1%). Eight out of ten (83.3%) physicians believed that palliative care is a hospital-based care (83.3%) only for older adults (74.5%) and cancer patients (63%). More than half (56.4%) of the respondents thought palliative care is exclusively for the last six months of life, and one-fifth (20.9%) were confused with the idea. The highest correctly scored item was "Palliative care is a team-based approach", and the lowest correctly scored item was "Palliative care is a hospital-based care".(Table 2).

The older physicians had significantly better knowledge than younger ones *(P = 0.01)*. Though the physicians with postgraduate trainings also have significantly better knowledge *(P = 0.01)*, nearly half (42.6%) of the physicians with graduate degree also had average knowledge level. The knowledge level also varied significantly, according to specialties *(P = 0.01)*. Nearly half of the respondents from each specialty had average knowledge. Still, four out of ten (44.5%) physicians from medicine and allied branches and five out of ten (48.1%) from gynecology and allied branches had excellent knowledge about palliative care (Table 3).

## Discussion

Palliative care is currently considered an integral part of medical care rather than a sub-specialty. Palliative care has also been recognized as a part of basic medical qualification by World Health Organization [24]. The current study reveals the preliminary results of the knowledge level of Bangladeshi physicians about this field.

**Table 2. Knowledge regarding palliative care according to PaCKs items (n = 479).**

| Variables | True n (%) | False n (%) | Don't Know n (%) |
|---|---|---|---|
| **General conceptions about Palliative Care** | | | |
| Team based approach | 438 (91.4) | 12 (2.5) | 29 (6.1) |
| Deals with psychological issues | 364 (76.0) | 72 (15.0) | 43 (9.0) |
| Helps better understanding of the treatment options | 340 (71.0) | 83 (17.3) | 56 (11.7) |
| Improves patient's daily activities | 421 (89.6) | 24 (5.0) | 35 (5.4) |
| Manages side effects of other treatments | 346 (72.2) | 78 (16.3) | 55 (11.5) |
| Addresses stress from serious illnesses | 405 (84.6) | 24 (5.0) | 50 (10.4) |
| Helps families to cope | 451 (94.2) | 11 (2.3) | 17 (3.5) |
| **Common misconceptions about palliative care** | | | |
| Exclusively for the last six months of life | 270 (56.4) | 109 (22.8) | 100 (20.9) |
| Only for the cancer patients | 302 (63.0) | 146 (30.5) | 31 (6.5) |
| Only for the older adults | 357 (74.5) | 88 (18.4) | 34 (7.1) |
| Only hospital-based care | 399 (83.3) | 52 (10.9) | 28 (5.8) |
| Encourages to stop taking curative treatments | 398 (83.1) | 46 (9.6) | 35 (7.3) |
| Encourages to stop consulting other specialties | 426 (88.9) | 24 (5.0) | 29 (6.1) |

Although the concept of palliative care is still new in Bangladesh, our study found that the majority (96.2%) of the physicians are familiar with this field without being directly associated. This situation is hopeful compared to the nonmedical science students, where only one-fourth of them are familiar with this term [16]. The situation is much improved than in other

**Table 3. Level of knowledge about palliative care based on demographic characteristics (n = 479).**

| Variables | Level of knowledge, n (%) | | | Test statistics | P-value |
|---|---|---|---|---|---|
| | **Poor** | **Average** | **Excellent** | | |
| **Sex***[***] | | | | | |
| Men | 79 (30.0) | 98 (37.3) | 86 (32.7) | 0.2 | 0.9 |
| Women | 60 (27.8) | 83 (38.4) | 73 (33.8) | | |
| **Age***[*] | | | | | |
| 20–30 | 115 (30.6) | 144 (38.3) | 117 (31.1) | 7.2 | **0.01** |
| 31–40 | 23 (23.5) | 37 (37.8) | 38 (38.8) | | |
| >40 | 1 (20.0) | 0 (0.0) | 4 (80.0) | | |
| **Qualifications***[***] | | | | | |
| Graduate | 93 (29.8) | 133 (42.6) | 86 (27.6) | 14.2 | **0.01** |
| Postgraduate or in course | 46 (27.5) | 48 (28.7) | 73 (43.7) | | |
| **Specialties***[**] | | | | | |
| Basic medical sciences | 15 (62.5) | 03 (12.5) | 06 (25.0) | 32.7 | **0.01** |
| Medicine and allied branches | 29 (22.7) | 42 (32.8) | 57 (44.5) | | |
| General practitioners | 40 (29.4) | 54 (39.7) | 42 (30.9) | | |
| Gynecology and Obstetrics | 5 (18.5) | 09 (33.3) | 13 (48.1) | | |
| Interns | 27 (36.0) | 32 (42.7) | 16 (21.3) | | |
| Surgery and allied branches | 15 (26.3) | 29 (50.9) | 13 (22.8) | | |
| Public health | 8 (25.0) | 12 (37.5) | 12 (37.5) | | |

[*]Fisher's Extract test;

[**]Monte Carlo extract test;

[***]Chi square test;

p value <0.05 considered as significant

countries like Pakistan and Vietnam, where only half of the medical professionals know this field [25, 26]. Aside from the familiarity with this field, the basic knowledge level of palliative care among the Bangladeshi physicians mostly (71%) ranged from average to excellent. This percentage is higher than the physicians and nurses with good or above knowledge about palliative care from countries like Pakistan (55.7%), Ethiopia (69.5%), Vietnam (64.9%), and Thailand (55.7%) where palliative care is also a new concept [25–27]. But the percentage is slightly lower than the neighboring country India (89.9%), with a relatively developed palliative care system [28]. In this study, most physicians without being directly associated with palliative care have fundamental knowledge about the general concepts and basic philosophy of palliative care, such as addressing psychological issues, including family members, coping, and team-based approaches. Overall, the situation casts a ray of hope for a country like Bangladesh, where palliative care is still developing and not included in the national health care delivery system or undergraduate medical curriculums. Patients do not directly seek palliative care here before consulting with other medical specialties. So, good knowledge about the fundamental concepts of palliative care among the physicians, especially the general physicians, might help recognize the need and delivery of this care to the patients from all specialties and subspecialties.

Our study also explored the prevailing common misconceptions about palliative care among physicians. The most prevailing misconceptions are that, palliative care discourages patients from consulting other specialties and refrains them from seeking curative treatment. This misconception is on par with the general people from a country with highly developed palliative care like United States [12]. This misconception may lead physicians not to advise their patients about seeking palliative care in the early state of the disease when curative treatment might be considered an option. These may lead to increased unnecessary treatment burdens and psychological and social sufferings and act as barriers to early access to palliative care.

Another widely prevailing misconception is about the target populations of palliative care. It is mostly believed that palliative care is only for the elderly or cancer patients and patients in the last six months of life. Different studies suggest that this misconception exists among health care professionals and general people worldwide [12, 18, 29]. Due to this misconception, physicians often fail to recognize many patients who need palliative care but do not belong to any of the groups mentioned earlier. This acts as a barrier to widespread access to palliative care. On that note, most physicians of Bangladesh consider palliative care as a hospital-based care, while home-based care is also an option. This leads to unnecessary hospitalization and hospital deaths which contradicts some philosophy of palliative care.

It is also noticeable from the study that knowledge level varies with the physicians' age and postgraduate training, most likely due to experience and a broader range of study. It is also evident that, among the younger physicians, the percentage of above-average basic knowledge about palliative care is comparatively low, indicating the less importance of this field at the undergraduate level. This situation is unique in Bangladesh and common among other countries like Pakistan, Vienna, and Brazil [24, 25, 30]. The excellent knowledge level in medicine and gynecology and their allied branches indicates that more emphasis is given on palliative care in this sector. Simultaneously, all specialties need to have a basic idea of palliative care because it is a multisectoral approach.

One limitation of this study is that it was conducted using only digital social media platforms. Although many physicians are active in these platforms, a vast group of physicians is left behind who do not use these platforms, especially older physicians. We have not included medical students who may not reflect the undergraduate level. There is a chance of non-response bias as the response rate of this study is very low (23.9%). Although it is close to an

average response rate of internet-based studies (20.4%) and both the respondents and non-respondents were from similar backgrounds, it does not eliminate the possibility of this bias [31]. Also, we did not have complete control over the respondents, so the true denominator of the outcome remains undetermined. Using only PaCKs questionnaire, which is a valid tool to assess palliative care's most fundamental knowledge level, it lets out some in-depth and widely discussed concepts like pain management, opioid use, breaking bad news, communication, the idea of death and dying, etc. This study would benefit from replication in a broad platform outside digital social media and including additional questions addressing the in-depth concepts of palliative care.

## Conclusion

In conclusion, this study found that although having fairly good knowledge about goals and basic concepts of palliative care, misconceptions are also highly prevalent in Bangladeshi physicians. These misconceptions act as a barrier to the development of this field. Our finding emphasizes the need to include the basic concept of palliative care in the medical curriculum and raise awareness among the physicians first to identify the need among target populations first.

## Supporting information

**S1 Appendix. PaCKs questionnaire.**
(DOCX)

## Author Contributions

**Conceptualization:** Jheelam Biswas.

**Data curation:** Jheelam Biswas, Palash Chandra Banik.

**Formal analysis:** Jheelam Biswas, Palash Chandra Banik.

**Investigation:** Jheelam Biswas.

**Methodology:** Jheelam Biswas, Palash Chandra Banik.

**Software:** Jheelam Biswas, Palash Chandra Banik.

**Supervision:** Palash Chandra Banik, Nezamuddin Ahmad.

**Visualization:** Jheelam Biswas.

**Writing – original draft:** Jheelam Biswas, Palash Chandra Banik.

**Writing – review & editing:** Palash Chandra Banik, Nezamuddin Ahmad.

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
