## [Decision Letter · Decision Letter 0]

26 Apr 2021

PONE-D-21-09118

Physicians' Knowledge about Palliative Care in Bangladesh: A cross-sectional study using digital social media platforms

PLOS ONE

Dear Dr. Biswas,

Thank you for submitting your manuscript to PLOS ONE. After careful consideration, we feel that it has merit but does not fully meet PLOS ONE’s publication criteria as it currently stands. Therefore, we invite you to submit a revised version of the manuscript that addresses the points raised during the review process.

The authors have attempted to describe the physicians' knowledge on Palliative Care in Bangladesh. However, there are certain major concerns regarding the article in its current form. The comments are appended below. Please address all the comments in a point-by-point manner.

We look forward to receiving your revised manuscript.

Kind regards,

Dr. Arista Lahiri, MBBS, MD

Academic Editor

PLOS ONE

Journal Requirements:

Additional Editor Comments:

The authors have conducted a study on the physician's knowledge on Palliative Care in Bangladesh. But there are certain issues that needs to be addressed apart from the reviewers' comments.

1. I am concerned about the representativeness of the sample. It appears that the authors did not have any control while collecting the data, because the dissemination was uncontrolled as it is commonly seen in social media-based studies. The main concern here thus lies regarding the denominator for the outcome.

2. The authors have reported a substantially high non-response. Were the respondents and non-respondents similar in terms of background characteristics? This information is important to justify the results given the non-response.

3. The sample size calculation should incorporate proper reference.

4. The authors need to provide information regarding reliability and validity of the study tool used. The authors themselves included a third option in the questionnaire as stated by them. This further warrants validation data on the modified tool. How did they handle this third option during analysis that also needs to be detailed.

5. The authors stated that this was a web-based survey. In this context how did they obtain informed consent from the participants?

6. The authors state that the physicians not involved with palliative care were included, that implies that those involved with palliative care were excluded. So no need to repeat what is written in the exclusion criteria, because again undergraduate students are not registered physicians. (P-4, L112-114). The authors need to elaborate on their selection criteria without repeating them.

Reviewers' comments:

Reviewer's Responses to Questions

**Comments to the Author**

1. Is the manuscript technically sound, and do the data support the conclusions?

Reviewer #1: No

Reviewer #2: Partly

2. Has the statistical analysis been performed appropriately and rigorously? 

Reviewer #1: No

Reviewer #2: I Don't Know

3. Have the authors made all data underlying the findings in their manuscript fully available?

Reviewer #1: Yes

Reviewer #2: No

4. Is the manuscript presented in an intelligible fashion and written in standard English?

Reviewer #1: Yes

Reviewer #2: No

5. Review Comments to the Author

Reviewer #1: Physicians' Knowledge about Palliative Care in Bangladesh: A cross-sectional study using digital social media platforms

Decision: The article is not suitable to be published in PLOS One.

Comments:

1. “The field of palliative care in Bangladesh is still in its infancy” – then naturally the physician’s knowledge about this care will be limited.

2. In most of the knowledge based study, it is found that the knowledge base is poor, which necessitates further intervention. Thus the research hypothesis is already known.

3. “The estimated sample size was 299 (considering familiarity with palliative care among Bangladeshi nonmedical students 75.2%).” – reference is missing.

4. “Knowledge level about palliative care was categorized into three categories following the Bell Curve theory by mean±1SD.” – did you check about the normality distribution of the data set? If data set is found to be skewed then there is no scope of applying this mean and SD concept. This statement is missing.

5. Table 1: Uniformity is not maintained while writing percentages. Although N (%) is written as caption then there is no necessity of writing –18(3.8%) in the last row.

6. Table 2: Footnote: what does this asterix means? Missing in the table.

Reviewer #2: Thank you for presenting this important topic in developing country setting. For improvement of the manuscript, I would like to suggest the following points.

1. In the “Background”, at line 79 the reference number becomes 20-22. I think it should be 13-15 according to the previous reference number. So you need to check the references in your text and update accordingly.

2. For description of your study’s aim at the end of “Background”, it would be better to write the respective text from line 93 to 101.

3. In “sample size and criteria”, for calculated number of participants, it would be good to put the reference in the text (line 112).

4. And I think the response rate can be reported in this section instead of reporting in the first part of “Result” (line 154-155).

5. For data analysis, I am not sure that one way ANOVA can be used for assessing the relationship between each item score and respective responses (line 144-5).

6. In “Results”, description about “Figure 1” (line 181), “Though seven out of ten (68.7%)…….”, could you please rewrite this sentence simply?

7. At line 199, 20.9% is not “one-fourth”. It would be “one-fifth”. I think it is spelling error.

8. For description of table 2, the comment is mentioned in comment 5.

9. For presentation of data in table 3, how about to omit the column of “Total”? Any more information cannot be given to the readers by putting it. Please mindful about the decimal place of p-values in table 3 (2 or decimal places).

6. PLOS authors have the option to publish the peer review history of their article (what does this mean?). If published, this will include your full peer review and any attached files.

Reviewer #1: **Yes: **Indranil Saha

Reviewer #2: No

---

## [Author Response · Author response to Decision Letter 0]

3 Jun 2021

Additional Editor’s comments to the authors-

Comments 1: I am concerned about the representativeness of the sample. It appears that the authors did not have any control while collecting the data, because the dissemination was uncontrolled as it is commonly seen in social media-based studies. The main concern here thus lies regarding the denominator for the outcome.

Reply 1: Thank you for your comments; we have revised the whole manuscript. As it was an internet based study we did not have any control regarding respondents. So we did not find the true denominator, which has been addressed in the limitation part of the Discussion. However we expressed our outcome as proportion in the result section instead of prevalence. The proportion of average to excellent knowledge level was found to be 71%. 

Changes in the text: 

• Discussion: 

Line 293-294: Also we did not have complete control over respondents, so the true denominator of the outcome remained undetermined.

• Result:

Line 189-190: The proportion of average to excellent knowledge about palliative care was 71%(Figure 1).

Comment 2: The authors have reported a substantially high non-response. Were the respondents and non-respondents similar in terms of background characteristics? This information is important to justify the results given the non-response.

Reply 2: Thank you for adressing the issue. The non-response rate is quite high which may lead to non response bias. In the revised manuscript we have discussed the issue in the limitation part of the discussion section. However the average resopnse rate in web based studies is about 20.4% (Deutskens E et al 2004, reference 31), and in our study it is quite close to the number(23.95%) Our respondents and non respondents were from similar backgrounds, however it does not completely eliminate the possiblity of non resonse bias. We have also adressed this in the sample size and criteria. 

Changes in the text: 

• Discussion:

Line 289-292: There is a chance of non response bias as the response rate of this study is very low (23.95%). Although it is close to average response rate of internet based studies (20.4%), as well as the respondents and non respondents were from similar backgrounds, but it doesn’t eliminate the possibility of this bias31 

• Methods: 

Sample size and criteria: 

Line 129: Initially, 2000 physicians who fulfilled the necessary inclusion criteria showed interest after learning about the study's objectives and purpose, but 479 returned with the complete response (response rate 23.95%).

Comment 3: The sample size calculation should incorporate proper reference.

Reply 3: Thank you. Proper reference has been added to the revised mauscript (reference no 21). 

Changes in the text: 

• Methods:

Sample size and criteria: 

Line 126-127: The estimated sample size was 299 (considering familiarity with palliative care among Bangladeshi nonmedical students 75.2%)[21]

• References: 

21. Pavel O, Ahmad N, Islam S. Assessing the Knowledge Pattern Regarding the Palliative Care among the Nonmedical Life Science Students in Bangladesh. J Life Sci. 2017;3(4):1110-1113

Comment 4: The authors need to provide information regarding reliability and validity of the study tool used. The authors themselves included a third option in the questionnaire as stated by them. This further warrants validation data on the modified tool. How did they handle this third option during analysis that also needs to be detailed.

Reply 4: Thank you for pointing out the issue. The modified version was adapted from a study done byKozlov E et al (2017), which has been mentioned with proper reference in the Data collection procedure part of the Methodology section of the revised manuscript. Also a pilot study was done among 50 physicians which yielded almost same resopnse. During scoring and analysis the total kowledge scores answers to 3rd option was considered and merged with the incorrect responses for each item, and scored as “0” as in the original scoring system which is mentioned in the Data analysis part of the revised manuscript.

Changes in the Text: 

• Methods: 

Data collection procedure: 

Line 138-140: This modified version was adapted from a study done by Kozlov E et al (2017) [12]. A pilot study was done for the purpose of validation of the tool among 50 physicians which yielded almost same result.

Data analysis: 

Line 152-153: The third option "I do not know" which was included to avoid guessing was also considered and merged with incorrect responses and scored “0”.

Comment 5: The authors stated that this was a web-based survey. In this context how did they obtain informed consent from the participants?

Reply 5: Thank you for mentioning the issue. The data was collected via Google form. An informed consent statement was attached at the beginning of the form as a mandatory field. The rest of the questionnaire was available and allowed to be submitted only after clicking “Yes” on the informed consent part. This issue has been mentioned in the Ethical consideration part of the revised manuscript and data collection procedure.

Changes in the text: 

Methods: 

• Ethical considerations: 

Line 171-174: An informed consent statement was attached at the 1st section of the Google form platform, which was a mandatory field. Interested participants were allowed to move to the next steps of the questionnaire and submit only after agreeing with the consent statement.

• Data collection procedure: 

Line 145-147: Data was collected via Google form platform and informed consent statement was added to the 1st Section as a mandatory field.

Comment 6: The authors state that the physicians not involved with palliative care were included, that implies that those involved with palliative care were excluded. So no need to repeat what is written in the exclusion criteria, because again undergraduate students are not registered physicians. (P-4, L112-114). The authors need to elaborate on their selection criteria without repeating them.

Reply 6: Thank you. The the exclusion and inclusion criteria was corrected in the revised manuscript.

Changes in the text:

• Methods: 

Sample size and criteria: 

Line 127-128: Registered physician from any discipline and not directly associated with palliative care was included in the study.

Reviewer#1’s comments to the authors-

Comment 1: “The field of palliative care in Bangladesh is still in its infancy” – then naturally the physician’s knowledge about this care will be limited.

Reply 1: Thank you. The line has been revised and changed in the revised manuscript

Changes in the text:

• Background:

Line 101: The field of palliative care in Bangladesh is still new and in the stage of continued development.

Comment 2: In most of the knowledge based study, it is found that the knowledge base is poor, which necessitates further intervention. Thus the research hypothesis is already known.

Reply 2: Thank you. However the aim of the study is to explore the current state of knowledge about palliative care among Bangladeshi physicians. There is no specific hypothesis in the study. As we found the though initial knowledge is average to excellent, but misconceptions are more prevent too.

Comment 3:“The estimated sample size was 299 (considering familiarity with palliative care among Bangladeshi nonmedical students 75.2%).” – reference is missing.

Reply 3: Thank you for pointing the issue. The reference has been added in the revised manuscript.

Changes in the text: 

• Methods:

Sample size and criteria: 

Line 126-127: The estimated sample size was 299 (considering familiarity with palliative care among Bangladeshi nonmedical students 75.2%)[21]

• Reference: 

21. Pavel O, Ahmad N, Islam S. Assessing the Knowledge Pattern Regarding the Palliative Care among the Nonmedical Life Science Students in Bangladesh. J Life Sci. 2017;3(4):1110-1113

Comment 4: “Knowledge level about palliative care was categorized into three categories following the Bell Curve theory by mean±1SD.” – did you check about the normality distribution of the data set? If data set is found to be skewed then there is no scope of applying this mean and SD concept. This statement is missing.

Reply 4: Thank you for pointing the issue out. We have reanalyzed the data set and found the distribution is skewed. So we re-catagorized the knowledge level according to inta quatrile range (IQR) and median.The value below the 25th quartile was categorized as poor, the range between 25th and 75th quartile was categorized as average, and the value above 75th quartile was categorized as excellent. Necessary changes made according to the new catagorization in the Table 3 and Figure 1, as well as in resultand in discussions.

Changes in the text: 

Methods:

Data analysis:

• Line 156: Descriptive analysis (frequency, percentage, median and intra quartile range) was done for categorical and quantitative variables. 

• Line 158-162: Knowledge level about palliative care was categorized into three categories according to the intra quartile ranges and median. The value below the 25th quartile was categorized as poor, the range between 25th and 75th quartile was categorized as average, and the value above 75th quartile was categorized as excellent Knowledge about palliative care.

Result: 

• Line 188-190: The Knowledge about palliative care score ranged from as low as 0 to the perfect score of 13, with the median score 11.0. The proportion of average to excellent knowledge about palliative care was 71% (Figure 1).

• Line 212-213: Though the physicians with postgraduate trainings also have significantly better knowledge (p=0.01), nearly half (42.6%) of the physicians with graduate degrees also had average knowledge level.

• Line 214-218: The knowledge level also varied significantly, according to specialties (p= 0.01). Nearly half of the respondents from each specialty had average Knowledge, but four out of ten (44.5%) physicians from medicine and allied branches and five out of ten (48.1%) from gynecology and allied branches had excellent Knowledge about palliative care (Table 3).

Discussions: 

• Line 235-239: Aside from the familiarity with this field, the basic knowledge level of palliative care among the Bangladeshi physicians mostly (71%) ranged from average to excellent. This percentage is higher than the physicians and nurses with good or above knowledge about palliative care from countries like Pakistan (55.7%), Ethiopia (69.5%), Vietnam (64.9%), and Thailand (55.7%) where palliative care is also a new concept[25-27]

• Line 274-276: It is also noticeable from the study that knowledge level varies with the physicians' age and postgraduate training, most likely due to experience and broader range of study.

• Line 280: The excellent knowledge level in medicine and gynecology and their allied branches indicates that more emphasis is given on palliative care in this sector.

Comment 5: Table 1: Uniformity is not maintained while writing percentages. Although N (%) is written as caption then there is no necessity of writing –18(3.8%) in the last row.

Reply 5: Thank you for pointing out the. The error in the Table 1 has been corrected in the revised manuscript.

Comment 6: Table 2: Footnote: what does this asterix means? Missing in the table.

Reply 6: Thank you for pointing out the error. The footnote has been deleted from table 2 in the revised manuscript. 

Reviewer#2 comments to the author-

Comment 1: In the “Background”, at line 79 the reference number becomes 20-22. I think it should be 13-15 according to the previous reference number. So you need to check the references in your text and update accordingly.

Reply 1: Thank you for pointing the issue out. The reference numbers has been updated in the revised manuscript.

Comment 2: For description of your study’s aim at the end of “Background”, it would be better to write the respective text from line 93 to 101. 

Reply 2: Thank you. The part regarding the aim of the study at end of the background has been revised and rewritten in the revised manuscript.

Changes in the text: 

Background:

Line 103-115: Very few studies were done to assess the knowledge level about palliative care in Bangladesh. A study among non-medical students of life science found that one-fourth of the participants are not aware of palliative care[21]. Another study found that only 10% of the practicing Bangladeshi physicians have knowledge about pain management in terminally ill patients[22]. However, no study has yet to provide a clear picture of the knowledge level or the prevailing misconceptions about palliative care among Bangladeshi physicians. As the demand of palliative care is rising, all physicians need to acquire proper knowledge about this field. Now it is high time to explore this knowledge gap. 

This paper aims to assess the knowledge level and explore the misconceptions about palliative care among Bangladeshi physicians to understand the situation better and raise awareness.

Comment 3: In “sample size and criteria”, for calculated number of participants, it would be good to put the reference in the text (line 112).

Reply 3: Thank you. The reference has been added in the revised manuscript.

Changes in the text: 

• Methods:

Sample size and criteria: 

Line 126-127: The estimated sample size was 299 (considering familiarity with palliative care among Bangladeshi nonmedical students 75.2%)[21]

• Reference: 

21. Pavel O, Ahmad N, Islam S. Assessing the Knowledge Pattern Regarding the Palliative Care among the Nonmedical Life Science Students in Bangladesh. J Life Sci. 2017;3(4):1110-1113

Comment 4: And I think the response rate can be reported in this section instead of reporting in the first part of “Result” (line 154-155).

Reply 4: Thank you. The response rate has been added to the end to “Sample size and criteria”as per your suggestion. 

Changes in the text: 

• Methods:

Sample size and criteria:

Line 129-131: Initially, 2000 physicians who fulfilled the necessary inclusion criteria showed interest after learning about the study's objectives and purpose, but 479 returned with the complete response (response rate 23.95%)

Comment 5: For data analysis, I am not sure that one way ANOVA can be used for assessing the relationship between each item score and respective responses (line 144-5)

Reply 5: Thank you. After reanalyzing the data, and found one way ANOVA in table 2 does not add any new significance in the result. So we excluded one way ANOVA from the revised manuscript form both table 2 and result section. 

Comment 6: In “Results”, description about “Figure 1” (line 181), “Though seven out of ten (68.7%)…….”, could you please rewrite this sentence simply?

Reply 6: Thank you. After reanalyzing the data, the sentence has been rewritten accordingly. 

Changes in the text:

• Result:

Line 189-190: The proportion of average to excellent knowledge about palliative care was 71%(Figure 1).

Comment 7: At line 199, 20.9% is not “one-fourth”. It would be “one-fifth”. I think it is spelling error.

Reply 7: Thank you. The error is corrected in the revised manuscript.

Changes in the text:

• Result:

Line 206: More than half (56.4%) of the respondents thought palliative care is exclusively for the last six months of life, and one-fifth (20.9%) were confused with the idea.

Comment 8: For description of table 2, the comment is mentioned in comment 5.

Reply 8: Thank you. The description of Table 2 has been corrected as in mentioned in reply 5.

Comment 9: For presentation of data in table 3, how about to omit the column of “Total”? Any more information cannot be given to the readers by putting it. Please mindful about the decimal place of p-values in table 3 (2 or decimal places).

Reply 9: Thank you. We have excluded the column ‘Total” in table 3, and corrected the p value issue. We have also updated the whole table after re-analyzing and re-catagorizing the data. 

Dear Reviewer, we are grateful for your kind time and substantial review; we believe now the manuscript is more improved, which will satisfy you.

---

## [Decision Letter · Decision Letter 1]

8 Jul 2021

PONE-D-21-09118R1

Physicians' knowledge about palliative care in Bangladesh: A cross-sectional study using digital social media platforms

PLOS ONE

Dear Dr. Biswas,

Thank you for submitting your manuscript to PLOS ONE. After careful consideration, we feel that it has merit but does not fully meet PLOS ONE’s publication criteria as it currently stands. Therefore, we invite you to submit a revised version of the manuscript that addresses the points raised during the review process.

Please consider incorporating the comments of reviewer 1. The revisions made are satisfactory.

We look forward to receiving your revised manuscript.

Kind regards,

Arista Lahiri

Academic Editor

PLOS ONE

Journal Requirements:

Additional Editor Comments (if provided):

Please consider incorporating comments of reviewer 1 in the manuscript.

Reviewers' comments:

Reviewer's Responses to Questions

**Comments to the Author**

1. If the authors have adequately addressed your comments raised in a previous round of review and you feel that this manuscript is now acceptable for publication, you may indicate that here to bypass the “Comments to the Author” section, enter your conflict of interest statement in the “Confidential to Editor” section, and submit your "Accept" recommendation.

Reviewer #1: All comments have been addressed

Reviewer #2: All comments have been addressed

2. Is the manuscript technically sound, and do the data support the conclusions?

Reviewer #1: Yes

Reviewer #2: Yes

3. Has the statistical analysis been performed appropriately and rigorously? 

Reviewer #1: Yes

Reviewer #2: Yes

4. Have the authors made all data underlying the findings in their manuscript fully available?

Reviewer #1: Yes

Reviewer #2: Yes

5. Is the manuscript presented in an intelligible fashion and written in standard English?

Reviewer #1: Yes

Reviewer #2: Yes

6. Review Comments to the Author

Reviewer #1: The article needs following clarifications:

1. Abstract: Introduction: Last line: Better to replace the word ‘explore’ by ‘assess’. Same corrections needed at the end of background.

2. Better to mention the year of the study in the abstract too

3. Table 3: Better to mention test statistics along with P value

Reviewer #2: The authors have addressed the reviewers' comments. I would like to give some minor comments for the accuracy of the manuscript.

1. For decimal points, you used 1 decimal points. So it would be great to use 1 decimal point for response rate in line 45 and line 131; mean age in line 177 and table 1.

2. Typing errors in 156 and 159 (it should be "interquartile range"); line 163 and 223 (the tests must be "Fisher's exact test, and the Monte Carlo exact test").

7. PLOS authors have the option to publish the peer review history of their article (what does this mean?). If published, this will include your full peer review and any attached files.

Reviewer #1: **Yes: **Indranil Saha

Reviewer #2: No

---

## [Author Response · Author response to Decision Letter 1]

12 Aug 2021

Additional Editor’s comments to the authors-

Comment: Please consider incorporating comments of reviewer 1 in the manuscript.

Reply: Thank you for your comments; we have revised the whole manuscript. We have considered and incorporated the revisions suggested by reviewer 1.

Reviewer#1’s comments to the authors-

Comment 1: Abstract: Introduction: Last line: Better to replace the word ‘explore’ by ‘assess’. Same corrections needed at the end of background.

Reply 1: Thank you. The line has been revised and changed in the revised manuscript

Changes in the text:

• Abstract : Introduction

Line 35: This study aims to assess the preliminary knowledge level and the misconceptions about this field among physicians.

• Background:

Line 112: This paper aims to assess the knowledge level and the misconceptions about palliative care among Bangladeshi physicians to understand the situation better and raise awareness.

Comment 2: Better to mention the year of the study in the abstract too

Reply 2: Thank you. The year of the study has been incorporated in the methods section of the abstract.

Changes in the text:

• Abstract : Methods

Line 38-41: This cross-sectional study was conducted among 479 physicians using a self-administered structured questionnaire adapted from Palliative Care Knowledge Scale (PaCKs) on various digital social media platforms from December 2019 to February 2020.

Comment 3: Table 3: Better to mention test statistics along with P value

Reply 3: Thank you for pointing the issue. The test statistics is mentioned in the Table 3 of revised manuscript in the column titled “Test statistics”. 

Reviewer#2 comments to the author- 

Comment 1: For decimal points, you used 1 decimal points. So it would be great to use 1 decimal point for response rate in line 45 and line 131; mean age in line 177 and table 1.

Reply 1: Thank you.The issue has been corrected in the revised manuscript body and also in Table 1.

Changes in the text: 

• Abstract: Results

Line 45: Almost equal number of physicians of both genders from four major specialties and their allied branches took part in the study (response rate 23.9%).

• Methods: Sample size and criteria: 

Line 128-130: Initially, 2000 physicians who fulfilled the necessary inclusion criteria showed interest after learning about the study's objectives and purpose, but 479 returned with the complete response (response rate 23.9%).

• Results: 

Line 176-177: Almost equal numbers of physicians from both genders took part in the study of mean age 28.9±3.6 years.

• Discussion:

Line 289-290: There is a chance of non-response bias as the response rate of this study is very low (23.9%).

Comment 2: Typing errors in 156 and 159 (it should be "interquartile range"); line 163 and 223 (the tests must be "Fisher's exact test, and the Monte Carlo exact test").. 

Reply 2: Thank you. The typing error has been corrected in the line 156. Regarding the name of the tests performed in the line 162 and 223, we have conducted 3 tests. For the variables ‘Sex’ and ‘Qualifications’ we have performed ‘Chi square test’; the variable ‘Age’ we performed ‘Fisher’s exact test’; and for the variable ‘Specialties’ we performed ‘Monte Carlo exact test’. The name of all three tests has been also corrected and mentioned in the Table 3 of the revised manuscript. 

Changes in the text: 

• Methods: Data analysis

Line 155-156: Knowledge level about palliative care was categorized into three categories according to the intraquartile ranges and median.

Line 162-164: Chi-square, Fisher's extract test, and the Monte Carlo extract test was done to compare the knowledge level with different study subjects' demographic variables setting the α level at 0.05. 

Line 223: *Fisher’s Extract test; **Monte Carlo extract test; ***Chi square test; p value <0.05 considered as significant

Journal requirement comments to the authors-

Comment: Please review your reference list to ensure that it is complete and correct. If you have cited papers that have been retracted, please include the rationale for doing so in the manuscript text, or remove these references and replace them with relevant current references. Any changes to the reference list should be mentioned in the rebuttal letter that accompanies your revised manuscript. If you need to cite a retracted article, indicate the article’s retracted status in the References list and also include a citation and full reference for the retraction notice.

Reply: Thank you. We have checked the reference list carefully and found the journal mentioned in the reference no 7 has been retracted. The retracted journal was “Merriman A. In the darkness of the shadow of death: a ray of hope: the story of Hospice Africa. J Palliat Care. 1993 Autumn;9(3):23-4. PMID: 8271102.”

So, we have replaced the above mentioned reference with an updated one. Both articles were conducted in almost same study setting and yielded the same conclusion. So we only updated the reference number 7 without any change in the text body. 

Changes in the text: 

• Reference: 

7. Grant L, Brown J, Leng M, Bettega N, Murray S. Palliative care making a difference in rural Uganda, Kenya and Malawi: three rapid evaluation field studies. BMC Palliat Care. 2011;10(1).

Dear Reviewers, we are grateful for your kind time and substantial review; we believe now the manuscript is more improved, which will satisfy you.

---

## [Editor Report · Decision Letter 2]

19 Aug 2021

Physicians' knowledge about palliative care in Bangladesh: A cross-sectional study using digital social media platforms

PONE-D-21-09118R2

Dear Dr. Biswas,

We’re pleased to inform you that your manuscript has been judged scientifically suitable for publication and will be formally accepted for publication once it meets all outstanding technical requirements.

Kind regards,

Arista Lahiri

Academic Editor

PLOS ONE
---

## [Editor Report · Acceptance letter]

23 Aug 2021

PONE-D-21-09118R2 

Physicians' knowledge about palliative care in Bangladesh: A cross-sectional study using digital social media platforms 

Dear Dr. Biswas:

I'm pleased to inform you that your manuscript has been deemed suitable for publication in PLOS ONE. Congratulations! Your manuscript is now with our production department. 

Kind regards, 

on behalf of

Dr. Arista Lahiri 

Academic Editor

PLOS ONE